# Could Protons and Carbon Ions Be the Silver Bullets Against Pancreatic Cancer?

**DOI:** 10.3390/ijms21134767

**Published:** 2020-07-04

**Authors:** Camille Huart, Jia-Wei Chen, Benjamin Le Calvé, Carine Michiels, Anne-Catherine Wéra

**Affiliations:** Unité de Recherche en Biologie Cellulaire (URBC), Namur Research Institute for Life Sciences (NARILIS), University of Namur, 5000 Namur, Belgium; camille.huart@unamur.be (C.H.); jia-wei.chen@unamur.be (J.-W.C.); benjamin.lecalve@unamur.be (B.L.C.); carine.michiels@unamur.be (C.M.)

**Keywords:** pancreatic cancer, protons, carbon ions, radioresistance, chemoresistance, targeted combination, immunotherapy

## Abstract

Pancreatic cancer is a very aggressive cancer type associated with one of the poorest prognostics. Despite several clinical trials to combine different types of therapies, none of them resulted in significant improvements for patient survival. Pancreatic cancers demonstrate a very broad panel of resistance mechanisms due to their biological properties but also their ability to remodel the tumour microenvironment. Radiotherapy is one of the most widely used treatments against cancer but, up to now, its impact remains limited in the context of pancreatic cancer. The modern era of radiotherapy proposes new approaches with increasing conformation but also more efficient effects on tumours in the case of charged particles. In this review, we highlight the interest in using charged particles in the context of pancreatic cancer therapy and the impact of this alternative to counteract resistance mechanisms.

## 1. Introduction

Due to late diagnosis and resistance to treatment, pancreatic cancer represents the fourth cause of cancer related deaths worldwide [1]. If outcomes are not improved, the disease is predicted to be the second leading cause of cancer mortality within the next decade [2]. Pancreatic ductal adenocarcinoma (PDAC) is the most frequent type of pancreatic cancer (>85%) associated with the highest mortality rate; the five-year survival does not exceed 7% [2,3]. Median survival rates are going from 25 months for the earliest stage, between 10 to 15 months for locally advanced to less than 5 months for metastatic cancer. For early stages of unmetastasised PDAC, surgery is the only treatment with curative intention. However, the majority of patients are diagnosed at advanced stages since the disease often presents itself with non-specific symptoms such as diffuse abdominal discomfort and loss of appetite. At the time of diagnosis, only 15% to 20% of pancreatic cancers are said to be operable, as the tumour is usually in the border of, or even encloses, important vessels such as the celiac artery, portal vein, hepatic artery or superior mesenteric vein/artery [3]. Unmetastasised PDAC are thus said to be: (1) resectable (R-PDAC), no tumour contact with the mentioned vessels; (2) borderline resectable (BR-PDAC) in case of venous involvement, which can be reconstructed after resection; (3) locally advanced (LA-PDAC) if there is an extended involvement of the vessels or if the reconstruction is not possible after the resection. Other parameters than anatomical consideration, such as tumour biology or patient condition, might be taken into account for this classification and to determine the best treatment option.

As of now, the main chemotherapeutic agents involved in the management of PDAC are FOLFIRINOX and gemcitabine-based drugs [4,5]. On the one hand, FOLFIRINOX consists of: (1) 5-fluorouracil (5-FU), an antimetabolite drug inhibiting DNA synthesis and folinic acid (leuvocorin) to potentiate the 5-FU anticancer activity; (2) irinotecan, a topoisomerase inhibitor inducing DNA strand breaks; (3) oxaliplatin, a platinum-based alkylating agent, which inhibits DNA repair and/or DNA synthesis. This combination, initially developed to treat metastatic colorectal cancer, was chosen in PDAC treatment for its synergism, differential mechanisms of action and non-overlapping toxicities of the drugs included in the combination [6]. On the other hand, gemcitabine exerts its anticancer properties by inhibiting DNA synthesis and thus blocking cell cycle progression. This drug is also used as chemotherapeutic agent in many cancers such as breast, ovarian and non-small cell lung cancers, especially paired with platinum-based drugs [7,8]. In PDAC, gemcitabine, associated with other chemotherapeutic agents, mostly 5-FU, capecitabine (an orally 5-FU prodrug) or nanoparticle-bound paclitaxel (nab-paclitaxel), is aimed to increase the response rate and survival benefits in patients [7,9]. Nab-paclitaxel is a novel albumin-bound, solvent-free and water-soluble formulation of paclitaxel, an anti-mitotic agent acting on the tubulins [10]. 

Radiation therapy can be combined to chemotherapy to increase local control. However, results of clinical trials failed to identify radiotherapy as an essential part of PDAC treatment [11]. In this review, we aim at putting forward the role that should be played by modern radiation therapy in the management of PDAC, especially when using charged particles (protons or carbon ions). We suggest an association of heavy charged particles with DNA repair targeted drugs as a very promising approach to act on local control and to enhance an immune response to trigger a systemic effect on pancreatic cancer.

## 2. Current Treatment for Unmetastasised PDAC

For R-PDAC and BR-PDAC, surgery is followed by adjuvant chemotherapy of gemcitabine plus capecitabine, a therapeutic approach generalised after the publication of the European Study Group for Pancreatic Cancer (ESPAC-4) trial in 2017. This study demonstrated the benefits of gemcitabine-capecitabine combination, with a five-year survival reaching 30%, compared with a gemcitabine monotherapy [5,9]. Currently, several studies for adjuvant therapy using nanoparticle nab-paclitaxel plus gemcitabine (APACT trial; NCT0196443) or modified FOLFIRINOX (mFOLFIRINOX) (PRODIGE trial; NCT01526135) are ongoing [5]. 

The place of radiation therapy (RT) combined to chemotherapy (chemoradiation, CRT) in adjuvant settings to surgery for the management of R-PDAC or BR-PDAC is still not settled due to conflicting results of clinical trials. Adjuvant CRT to surgery, namely a total of 40 Gy delivered in daily 2 Gy fractions associated with 5-FU, led to survival benefits in GISTG [12] and EORTC trials [13], while a decrease in the median survival was observed in ESPAC-1 [14]. The RTOG 9704 trial focused on comparing 5-FU and gemcitabine combined to RT [15]. In this trial, less than 30% local recurrences were observed, namely half compared to previous trials, but more than 70% presented distant relapse. Interestingly, RTOG 9704 included a quality assurance for radiation plan and delivery which allowed a secondary analysis highlighting that deviation from the specified RT protocol guidelines had deleterious impact on survival [16]. This is now corrected in the RTOG 0848 trial with a “real time, prospective, mandated review and correction of deviations prior to the start of radiotherapy” as stated in the aforementioned paper. This last study demonstrates the high constraint in dose conformation and RT regimen for the treatment of PDAC. 

## 3. From Adjuvant to Neoadjuvant Treatment For Unmetastasised PDAC

After resection, margin analysis is of prime importance. In a retrospective study, Konstantinidis et al. showed that R1 resected patients, i.e., with at least one margin infiltrated with cancer cells, had only a slightly improved median survival compared to patients with LA-PDAC. Moreover, for a notable increase in the median survival, the resections without any trace of cancer cells, R0, had to be wider that 1 mm (35 vs. 16 months if margins stay within 1 mm) [17]. This shows that the reduction of margin positivity is required if one hopes to increase the overall survival of resected patients. 

In their recent review, Hall and Goodman gathered data from different trials and studies and evidenced that neoadjuvant CRT is superior to adjuvant chemotherapy when considering: the rate of positive margins (2–20% vs. 16–60%), the incidence of node positivity (17–40% vs. 62–80%) and the rate of local recurrence (5–15% vs. 19–53%) [11]. 

Recently, the PREOPANC trial compared preoperative CRT (36 Gy in 15 fractions + gemcitabine) to upfront surgery on R- and BR-PDAC. Among the assessed endpoints, the median survival was extended for BR-PDAC, the loco-regional and distant metastasis free interval were superior as well as the R0 rate (especially for BR-PDAC, 79% vs. 13%) [18]. According to multiple studies and analyses, FOLFORINOX seems to be the most effective regimen for neoadjuvant therapy in pancreatic cancer. However, not all patients benefit from this multi-drug regimen as it causes considerable toxicity [6,19,20]. The use of nab-paclitaxel paired with gemcitabine is also suggested to be a good strategy for neoadjuvant therapy [10,21].

Besides increasing local control for R-PDAC and BR-PDAC, neoadjuvant C(R)T can lead to secondary resectability of LA-PDAC thanks to the shrinkage, or even down-staging, of the tumour [22]. In their meta-analysis of responses to preoperative/neoadjuvant therapies, Gillen et al. conclude that one third of the patients that were initially considered non-resectable would benefit from surgery after neoadjuvant treatment at survival rate similar to initially resectable patients [23].

With the rise of more effective chemotherapies regarding systemic control, loco-regional control will become essential and RT is expected to play its part. With radiation therapy being a rapidly evolving field with new techniques in beam delivery, imaging capabilities and treatment planification algorithms, the next trials should associate modern techniques to take full advantage of RT and evaluate the benefit of CRT as (neo)adjuvant treatment of pancreatic cancer, and especially for LA-PDAC.

## 4. Pushing Forward Loco-Regional Control: Modern RT

Therapies aim to reach tumour control while minimising as much as possible normal tissue complications. This is the so-called therapeutic window. In the case of pancreatic cancer, the poor outcome achieved today shows that this window is quite small. While current treatments face resistance, dose escalation of CT or RT is limited by toxicities. New development in RT aims to reduce induced toxicities to allow dose escalation and hypofractionation. In addition, charged particles could avoid some of the resistance mechanisms present in PDAC. 

### 4.1. Dose Conformation Towards Dose Escalation and Hypofractionation

The anatomical position of the pancreas implies a highly conformational dose deposition to reach a sufficient coverage of the tumour volume while sparing surrounding healthy tissues and organs at risk (OARs). Evolution in dose conformation of conventional photon beams goes from two-dimensional (2D) delivery for early trials such as GIST to 3DCRT (three-dimensional (3D) conformation radiotherapy) and IMRT (intensity modulated radiation therapy) for RTOG 0848. Studies show that IMRT should be favoured, as it leads to a significant reduction in toxicities [24]. Moreover, the sparing of OARs with a better dose conformation allows a hypofractionated regimen, associated with a higher radiobiological response (biological dose equivalent, BED) but also with a reduced overall time of treatment. Even ablative irradiation with BED superior to 100 Gy can be considered in the case of SBRT (stereotactic body radiotherapy) with image guidance. 

SBRT is usually indicated in case of tumours with minimum motion uncertainty (brain, spinal cord) or for small tumours in organ with parallel functional subunits (lung, liver). This is not the case for pancreatic cancer as it is close to the gastrointestinal tract (serial functional subunits) but is also associated with uncertainties due to respiratory motion and luminal organ inter-fraction shape changes. For these reasons, SBRT treatments for pancreatic cancer need to be very carefully planned and delivered and associated with motion management. Krishnan et al. reported an improved overall survival (OS) (17.8 months vs. 15.0 months) for LA-PDAC patient receiving a BED higher than 70 Gy after chemotherapy treatment [25]. At the time of the publication, this was associated with expected survival rate of 36% and 31% at two and three years, respectively. Based on this study, two clinical trials for LA-PDAC, currently recruiting, will study hypofractionated ablative IMRT (NCT03523312) and the use of stereotactic magnetic resonance guided adaption radiation therapy (SMART, NCT03621644) based on very encouraging preliminary results [26].

In the seeking of dose conformation, charged particles, such as protons or carbon ions, present a clear advantage compare to photons thanks to their depth dose profile. Photons deposit most of their energy close to the surface entrance followed by a continuous decrease characteristic of their attenuation. This leads to dose deposition upstream and downstream the tumour. Charged particles, however, deposit a small fraction of their energy before what is called the Bragg peak, characterising the maximal energy released when the particles come at rest. The dose sharply decreases beyond this peak, allowing to spare downstream tissues. The position of the Bragg peak can be tuned to coincide with the tumour position and highly conformal dose deposition profiles can be obtained. Since the early 2000s, the introduction of beam scanning methods led to further improvement of conformation. As shown by Ling et al. in the frame of RTOG-0848, IMRT does slightly better than 3DCRT, but proton therapy (PT) outperforms both, with a clear reduction of deposited dose to OARs [27]. In their review, Rutenberg and Nichols concluded that PT is effective before or after surgical resection of PDAC. Moreover, as PT is well tolerated and reduced toxicities, it allows either further dose escalation or intensification of chemotherapy without delaying surgery [28]. Kim et al. demonstrated that adjuvant chemotherapy (capecitabine or 5-FU) could be safely administrated to PT and improve the OS [29]. In another study, Hiroshima et al. treated LA-PDAC patients with concurrent chemoradiotherapy with protons at dose of 50, 54–60 and 67.5 GyE [30]. They showed that OS was largely improved with increasing dose with 13.1 months at 50 GyE to 28.4 months at doses between 54–60 GyE and up to 42.5 months at 67.5 GyE. Moreover, the local recurrence time was higher than 36 months for doses from 54 GyE. Further developments in proton beam delivery are undertaken with, notably, continuous spot-scanning proton arc (SPArc), which can further decrease dose to OARs [31,32]. Shinoto et al. studied the concurrent gemcitabine with carbon ions for LA-PDAC patients [33]. They found that full dose of gemcitabine (1000 mg/m²) was safely administrated with 55.2 GyE carbon therapy. At two years, the freedom from local progression rate was 83% and OS rates equal to 48%. More results of clinical trials based on SRBT, proton beams and carbon ions are summarised in [34]. Table 1 presents active and recruiting clinical trials for the treatment of pancreatic cancer with protons or carbon ions.

A last point worth noting is the development of FLASH irradiation, i.e., irradiation at ultrahigh dose-rate (>40–100 Gy/s compared to <0.1 Gy/s for conventional RT). Studies have observed that normal tissues irradiated at such a dose-rate display reduced toxicities compared to irradiation at the same dose with conventional RT, this is called the “FLASH effect” (reviewed in [35,36]). This effect was reproduced by different teams in a wide range of animal models or organs and even with different types of radiation (X-rays, electrons or protons). They reported an increased tolerance of normal tissues to FLASH RT with reduction in acute and delayed toxicities such as pneumonitis, lung fibrosis, cognitive impairment, skin necrosis etc. [35,36]. Moreover, as explained in these reviews, not only does FLASH RT reduce tissue toxicities, it maintains, and sometimes even increases, the response of tumour cells. On the one hand, FLASH RT can be used to reduce normal tissue complication at an equivalent tumour control; on the other hand, FLASH RT allows dose escalation to achieve tumour control at the same toxicity level. This is shown by Favaudon et al., who irradiated C57BL/6J murine whole lungs as well as xenografted human tumours and syngeneic orthotopic lung tumours with conventional and FLASH RT [37]. They demonstrated that FLASH RT protected the lungs from radiation-induced fibrosis as well as the blood vessels and bronchi from radiation-induced apoptosis but led to the same tumour control as the one obtained by conventional RT. Moreover, as FLASH RT allowed dose escalation, a better tumour control was achieved with FLASH at increased dose. This was also demonstrated with FLASH proton (FLASH PT) irradiation in C57BL/6J mice, presenting or not pancreatic flank tumour derived from KPC (LSL-Kras^G12D/+^;LSL-Trp53^R172H/+^;Pdx-1-Cre) [38]. Toxicities to normal tissue were reduced after FLASH PT compared to standard 15 Gy PT irradiation of the whole abdomen. Reduced intestinal fibrosis was also observed with FLASH regimen while maintaining the same tumour control as standard PT delivery. The first patient, who presented a multiresistant CD30+ T-cell cutaneous lymphoma, was treated with FLASH RT (electron beam) at the Lausanne University Hospital in 2019 with very encouraging complete tumour response with a short follow-up of five months [39]. 

### 4.2. Charged Particles to Increase Dose Response and Counteract PDAC Resistance to Treatment

Chemo- and radio-therapies are subjected to resistance mechanisms developed by tumour cells and PDAC is no exception. Some of these mechanisms could be counteracted by protons or carbon ions. Indeed, as explained below, the DNA damage distribution is not homogeneous but rather clustered after charged particles irradiation while this pattern is homogeneous with chemotherapeutic drugs and conventional RT. These more complex damages are less easily repaired, hence, leading to enhanced cell death.

#### 4.2.1. Mechanisms of Resistance

Resistance can be intrinsic (de novo) and/or acquired in response to challenges during the treatment. Mechanisms of drug resistance in PDAC include the presence of highly resistant cancer cells, up- or down-regulated expression of specific microRNAs, aberrant gene expression, mutations and deregulation of key signalling pathways as well as features of the microenvironment and its components. 

Intrinsic resistance is associated with proteins undergoing mutations during PDAC development, mostly; oncogenic with KRAS mutations (observed in more than 90% of patients) and tumour suppressor mutations such as p53, CDKN2A or SMAD4 mutated in more than 50% of patients [3,40]. 

Besides its role in sustained growth of PDAC, KRAS activation leads to a metabolic reprogramming that gives a selection advantage to the transformed cells, as PDAC tumours are subject to high metabolic stress due to severe hypoxia and limited nutriment availability [3]. Autophagy, glycolysis, glutamine uptake and NRF2 antioxidant program are particularly active in KRAS mutated PDAC [3]. Recently, it was shown that KRAS^G12C^ could be targeted efficiently in pre-clinical and clinical settings. Indeed, treatment with ARS-1620, a KRAS^G12C^ inhibitor, demonstrated significant tumour growth inhibition as well regression in a patient-derived xenograft (PDX) mouse model of PDAC [41]. The inhibition of KRAS^G12C^ by AMG-510 was also found successful in a phase I trial in which patients presenting non-small cell lung cancers or colorectal cancers were enrolled. Out of the 22 patients enrolled, six patients achieved stable disease and one patient reached partial response [42]. Unfortunately, mutations in G12C only represent 1–4% of all KRAS mutations. Indeed, the predominant KRAS mutations in PDAC are KRAS^G12D^ and KRAS^G12V^, which are currently undruggable. Since directly targeting KRAS has proven difficult, therapies targeting the major downstream effector pathways are currently developed. These effectors include the RAS–RAF–MEK–ERK and PI3K–PDPK1–AKT signalling pathways. Several MEK inhibitors proceeded to clinical trials; however, they were shown to be inefficient in patients with PDAC as demonstrated in NCT01016483 and NCT01231581 trials. On the other hand, combining a MEK inhibitor to an autophagy inhibitor could display anti-proliferative effects as shown in cell lines and PDAC PDX tumours in mice [43]. 

Additionally to metabolism reprogramming, NRF2 has been shown to promote proliferation, angiogenesis, resistance to apoptosis and, through the epithelial to mesenchymal transition (EMT) activation, metastasis [44]. More details are awaited; however, Bailleul et al. did observe that NRF2 promotes radioresistance thanks to a powerful antioxidant response through metabolic reprogramming and pro-survival autophagy [45]. Autophagy is associated with either pro-survival or pro-death mechanisms depending on the circumstances [40,46]. Moreover, Wang et al. showed that SMAD4 deletion induces radioresistance through elevated reactive oxygen species (ROS) production along with autophagy induction [47]. These studies highlight that PDAC is able to manage ROS, with increased or decreased production, to promote resistance. 

Mutations in cyclin-dependent kinase inhibitor 2A (CDKN2A) were shown to be often present in PDAC patients [48]. CDKN2A is known to code for two tumour suppressor proteins, p16INK4A (inhibitor of CDK4/6) and p14ARF (activator of p53). Thus, mutations in CDKN2A are associated with an increase in cellular proliferation. For this reason, two phase I clinical trials are currently investigating the use of CDK4/6 inhibitors in combination with mTOR inhibitors or nab-paclitaxel in patients with PDAC (NCT03065062 and NCT02501902, respectively). Furthermore, it was demonstrated that sequential administration of CDK4/6 inhibitors after taxane treatment resulted in a significant reduction in the proliferation of PDAC cells in vitro and in vivo. CDK4/6 inhibitors prevented the cells treated with taxane to re-enter the cell cycle but also repressed homologous recombination [49]. Recently, encouraging pre-clinical results were obtained by combining a CDK4/6 inhibitor to MEK inhibitor [50]. Indeed, it was shown that combined treatment with a MEK inhibitor (trametinib) and a CDK4/6 inhibitor (palbociclib) induced a senescence-associated secretory phenotype (SASP), which led to a vascular remodelling in two mouse models of PDAC. This SASP-mediated vascular remodelling increased blood vessel density and permeability making the tumour more sensitive to the cytotoxic chemotherapy such as gemcitabine. These studies provide a strong proof of concept to combine CDK4/6 inhibitors to available chemotherapies in pancreatic cancer in order to amplify the effects of the latter.

Furthermore, pancreatic tumours are characterised by an abundant and dense collagenous stroma resulting from desmoplasia. The desmoplastic reaction observed in PDAC can be recognised through an excessive production of extra-cellular matrix with abundance of fibrillar proteins, glycoproteins, proliferating fibroblasts, inflammatory cytokines and immune cells [2,51]. The replacement of normal parenchyma with a desmoplastic environment increases pressure and causes vasculature constriction impairing drug delivery while creating a hypoxic environment for cancer cells [52]. 

Two types of hypoxia have been described: chronic hypoxia and cycling hypoxia. Chronic hypoxia corresponds to a deficit in oxygen for a continuous period of time, while cycling hypoxia consists of a back and forth between deep hypoxia and moderate hypoxia. Chronic and cycling hypoxia have different effects on resistance to treatment, tumour angiogenesis and tumour metastasis. Indeed, chronic hypoxia is going to severely influence the blood vessels while cycling hypoxia is known to affect cancer and stromal cells (for a review, see [53]). Tumour sensitivity to radiations depends on the degree of hypoxia at the time of irradiation and the duration of exposure to hypoxic conditions. However, studies have reported a higher radioresistance of cells exposed to cycling hypoxic conditions compared to the ones exposed to chronic hypoxia [54,55].

Hypoxia initiates the stabilisation of the Hypoxia Inducible Factor 1 (HIF-1α), which is one of the two subunits of the HIF-1 transcription factor. HIF-1 is key factor of many different signalling pathways implicated in tumour progression as well as in radio- and chemoresistance to cancer treatment [56,57]. In PDAC, Yokoi and Fidler demonstrated that hypoxia was able to mediate resistance to gemcitabine-induced apoptosis in the metastatic L3.6pl human pancreatic cancer cell line. This anti-apoptotic action was mainly exerted through the PI3K/Akt/NF-kB pathway, which is hyperactive under hypoxic conditions [58]. The hypoxia-induced chemoresistance also seems to act through the regulation of efflux pumps expression such as the ATP-binding cassette super-family G member 2 (ABCG2), which is a transmembrane protein responsible for the export of several chemotherapy drugs. In a study in the pancreatic cancer Capan-2 cell line, hypoxic conditions induced the overexpression of ABCG2 and resistance to gemcitabine [59]. This phenotype was suggested to be mediated by the phosphorylation of ERK1/2 which then activates HIF-1α in hypoxia. The activation of HIF-1α in turn regulates ABCG2 transcription by binding to its gene promoter. The inhibition of ERK1/2 and HIF-1α resulted in increased gemcitabine sensitivity in the Capan-2 cells [59]. Regarding radiotherapy, decreasing HIF-1 expression by HIF-1α inhibitor PX-478 was able to radiosensitise Panc-1 cells and BxPC-3 cells to X-ray irradiation in vitro [60]. In vivo, PX-478 markedly potentiated the anti-tumour activity of fractionated irradiation treatment, combined or not to 5-FU or gemcitabine chemotherapy, in Panc-1, CF-PAC-1 or SU.86.86 xenograft mice models [60]. Hypoxia also induces several changes in cancer cell metabolism that results in the physiochemical changes in the tumour microenvironment such as reduced pH and increased production of ROS [51]. Overall, there is strong evidence supporting the role of hypoxia and HIF-1 in treatment resistance in PDAC. 

Radiotherapy used to be given at relatively small doses over several weeks, since the normal tissues present a slight survival advantage compared to the tumours. Importantly, this type of treatment regimen allows for the reoxygenation phenomenon to occur. Indeed, hypoxic tumour cells become oxygenated by the time of the next treatment [61]. As previously mentioned, recent advances in the field of radiotherapy have led to an increased usage of SBRT, which delivers high doses in fewer fractions. In the case of PDAC, in which hypoxia can significantly vary, SBRT might not be optimal, depending on the extent to which reoxygenation can occur. However, as explained by Nahum in his work, for tumours associated with a lower alpha-beta ratio (parameters of the Linear-Quadratic model fitting survival fraction curves), a high number of fraction is not expected to increase tumour control for a given healthy tissue toxicity [62]. Pancreatic cancer is considered to have an alpha-beta ratio around 6 Gy, which is lower than the 10 Gy usually used for tumour cells [63]. This shows that PDAC could benefit from SBRT and the presence of hypoxia would then indicate that a higher overall dose should be used compared to oxic tumours. It also needs to be noted that, unexpectedly, high local controls have been observed following SBRT treatment [64]. The tumour response mechanisms behind high single dose efficacy are not fully understood and are still a work in progress, especially regarding the role of the tumour microenvironment [65,66].

Cancer stem cells (CSCs) and epithelial to mesenchymal transition (EMT) play a critical role in resistance to treatment [67,68,69]. CSCs are characterised by their capacity to self-renew and to generate multiple cell types along with cell division and tumour expansion. In PDAC, these cells expressing markers such as CD44, CD24, epithelial-specific antigen, CD133, CXCR4 or aldehyde dehydrogenase can also be used as potential CSCs markers [70,71,72,73]. EMT is a pro-invasive gene expression and signalling program enabling cancer cells to decrease cell-to-cell adhesion or cell-to-extracellular matrix and to acquire expression of mesenchymal proteins. During EMT, cancer cells undergo morphological changes and reorganise the cytoskeleton, resulting in increased motility and invasion abilities [74]. Besides its role in metastasis, a correlation between an EMT phenotype and chemo- or radioresistance has been established in cancer cells and notably in PDAC [69,75,76,77]. Several pathways, such as TGFβ, Wnt, Notch, EGFR, ERK or PI3K/AKT, are activated by RT or CT and promote the EMT phenotype. Subsequently, EMT can trigger transcription factors implicated in resistance (Snail, ZEB or Twist superfamily) [68,69,72,77,78,79]. Some of these pathways or regulators overlap with the ones activated by CSCs in PDAC such as TGFβ [80], Slug (Snail2) [81,82] or Notch1 [68]. 

CSC resistance to drug might also be explained by their relative quiescence state, i.e., a non-proliferative state. Quiescent cells are thought to be protected from chemotherapeutics agents as most of the drugs mediate their effects on one of the major characteristics that differentiate normal cells from cancer cells, which is their ability to indefinitely proliferate [83,84]. In PDAC, Cioffi et al. evidenced a quiescence-mediated chemoresistance in primary human pancreatic cancer cells obtained from patients and expanded in PDX [85]. The gemcitabine-resistant CSCs, isolated from PDX, were characterised by low expression of the miR-17-92 cluster members. Overexpression of miR-17-92 cluster in CSCs resulted in loss of the stemness phenotype, and a reduction of cells residing in G0 phase and G1 phase along with the abrogation of chemoresistance to gemcitabine. Inversely, the inhibition of the miR-17-92 cluster in non-CSCs PDAC cells induced a gain of stem-like features, abrogated cell proliferation and, subsequently, increased chemoresistance to gemcitabine [85]. 

Hypoxia-induced treatment resistance also seems to interplay with mechanisms implicating CSCs and EMT. For example, a link has been made between hypoxia-induced EMT and resistance to gemcitabine in PDAC cells [86]. Under hypoxic conditions, the PANC-1 and BxPC3 cell lines displayed an EMT-like phenotype, HIF-1α and NF-κB hyperactivity and reduced sensitivity to gemcitabine. The inhibition of the two transcription factors HIF-1α and NF-κB with siRNAs resulted in the reversal of EMT phenotype and an increased sensitivity of both cell lines to gemcitabine [86]. Furthermore, hypoxia was also shown to synergistically enhance gemcitabine-induced stemness and acquired resistance in Panc-1 and Patu8988 pancreatic cancer cell lines by activating the AKT/Notch1 signalling cascade, which evidenced that hypoxia plays an important role in maintaining and promoting CSC subpopulation in the tumour [87].

MicroRNAs (miRNAs) are non-coding small RNAs that are 17–25 nucleotides long. Some of them play an important role in drug sensitivity regulation in cancer cells. They act at a post-transcriptional level by binding mainly to 3′ untranslated region of target mRNA, which leads to repression or degradation of the mRNA [68]. MiRNAs can either act as oncogenes or tumour suppressors and their expression level was also reported to be correlated to drug and radiation response during cancer treatment [88,89,90]. For example, high levels of miRNA-21, miRNA-1266 and miR-221 were reported to be linked with gemcitabine resistance in PDAC [91,92,93,94]. On the contrary, miRNA-101-3p seems to be lowly expressed in gemcitabine-resistant pancreatic cancer cells and of miRNA-101-3p mimic transfection restored cell sensitivity for the drug [95]. Much evidence also suggests the implication of miRNAs in chemosensitivity through the regulation of EMT process and stemness in cancer cells. For instance, the expression of some of miR-200 and let-7 miRNA family members is downregulated in gemcitabine-resistant cells with an EMT phenotype. The re-expression of miR-200 by transfection in gemcitabine-resistant cells resulted in a morphological reversal of the EMT phenotype and increased sensitivity to gemcitabine [96]. Compared to normal pancreatic ductal epithelial cells, miR-34 expression is reduced in pancreatic CSCs (CD44+/CD24+/ESA+) and pancreatic cancer tumour cells (MIA PaCa-2 and AsPC-1). The restoration of miR-34 expression inhibits growth and enhances sensitivity to gemcitabine [97]. Furthermore, miRNA-320a and miRNA-221-3p were found to be upregulated in 5-FU-resistant pancreatic cancer cells [98,99,100]. As demonstrated by deep sequencing, upregulated and downregulated miRNAs are implicated in radiation resistance of PDAC cell lines [101]. For example, miRNA-216a and miRNA-23b were also found to be downregulated in radioresistant PDAC cell lines [102,103]. Both studies pointed to a pro-survival role of autophagy after radiation as the re-expression of both miRNAs inhibited autophagy and increased cell death. 

Growing evidence has brought the implication of extracellular microvesicles (EVs) as factors in chemo- and radioresistance of cancer cells [104,105,106,107]. EVs are microparticles with a lipid bilayer membrane, secreted from all cell types either in physiological and pathological conditions [81]. EVs are able to mediate drug resistance but also to confer resistance to drug-sensitive cancer cells through the transfer of cargoes including drug efflux pumps, pro-survival factors and inhibitors of apoptosis [104]. A part of EVs-mediated drug-resistance also relies on miRNAs transfer, as it was shown for miR-155 in gemcitabine-resistant pancreatic cancer cells [108]. Furthermore, these EVs containing miRNA can also be derived from immune cells such as tumour-associated macrophages (TAMs) or from cancer associated fibroblasts (CAFs), evidencing the existence of a complex collaboration between stromal or immune cells and cancer cells [104,109,110].

#### 4.2.2. Charged Particles vs. Resistance

In addition to dose conformation, charged particles have a higher linear energy transfer (LET), i.e., an increased ionisation density, especially within the Bragg peak. The LET varies along the particles track, being small in the entrance region and increasing rapidly within the Bragg peak. The same profile is observed for the density of induced damages, i.e., smaller when passing through the body and much higher within the tumour. Increased ionisation leads to the formation of more complex DNA damages (clustered lesions) than photons and thus to a stronger cellular response [111,112]. Radiation-induced DNA damages are classified as direct damages (ionising radiation interacts directly with the DNA) and indirect damages (DNA damage is mediated through ROS). With increasing LET, and thus increasing ionisation density, direct damage proportion increases as well as the complexity of the DNA damage pattern and the release of DNA fragments [113,114,115]. This leads to an increased cell response, which is described by the relative biological effectiveness (RBE). The RBE is determined as the ratio of the doses required to obtain a given output for a reference radiation (photons) compared to the chosen radiation [111]. For a given particle, the RBE has been found to depend on several factors such as the dose and dose-rate or the intrinsic radiosensitivity of the tissue [112]. For protons that have an LET close to photons at high energies, a 1.1 RBE value is used in clinic, although within the Bragg peak this value increases. For carbon ions, an RBE of 3 is used, reflecting the higher complexity of the DNA damage pattern. 

Thanks to this change in damage distribution, the use of charged particles such as protons or carbon ions might help to counteract some of the resistance mechanisms observed during cancer treatment [116,117].

For example, as high LET particles increase the production of direct DNA damages, their effect relies less on indirect damage mediated by ROS. As demonstrated by Georgakilas et al., with increasing LET, the number of induced cluster of DNA damage under normoxic and anoxic condition tends to be similar while it decreases for low LET photons [118]. This reduced addiction to ROS to induce DNA damages allows high LET particles to treat more efficiently hypoxic tumour [119,120,121]. Furthermore, photons and charged particles differ regarding the effect on signalling pathways: photons might upregulate one pathway while charged particles might induce the opposite effect, i.e., downregulate the very same pathway. For example, in glioblastoma, a differential effect of carbon ions versus photons was observed on orthotopic, syngeneic murine xenografts as well as glioma stem cell-enriched, PDX [122]. In their work, contrarily to photons, carbon ions downregulated Notch and Wnt pathways, angiogenesis, EMT and extracellular matrix remodelling. Works on CSCs response after charged particle irradiation, summarised in [116,123,124], showed that CSCs are more sensitive to protons and carbon ions than to photons. Moreover, inversely to conventional RT, HIF-1α has been found to be downregulated after proton or carbon ion irradiation [125,126]. The clustered DNA pattern and more particularly ROS distribution was hypothesised to be an explanation for the differences in gene upregulation or downregulation after conventional irradiation versus proton or carbon ion irradiation [126]. ROS distribution is homogeneous following photon exposure while it is concentrated around the ion track with charged particles [126].

Regarding PDAC, interesting results have been obtained in PDAC CSCs after carbon ion irradiation. Oonishis et al. analysed colony, spheroid and tumour formation as well as DNA double strand break (DSB) formation on PDAC CSCs treated with X-ray or carbon ions [127]. The proportion of CSCs was more enriched after X-rays compared to carbon ion irradiation. This was associated with an increased complexity of DSBs in the case of carbon ion irradiation. Similarly, Sai et al. showed that the proportion of CSCs was enriched after exposure to photons compared to carbon ions [128]. They evidenced that combination of carbon ions with gemcitabine synergistically enhanced CSCs death compared to carbon alone through an increase in complex DNA damage, in cell death (apoptosis and autophagy) and inhibition of cell proliferation. 

Additionally, high LET radiation can induce apoptosis in a p53-independent manner thanks to the activation of caspase-9 instead of caspase-8 apoptotic pathway [129,130,131]. This feature is interesting knowing that TP53 is mutated in 60% to 70% of PDAC patients [3]. To target other mutations often found in PDAC, Ruscetti et al. used inhibitor of CDK4/6 and MEK, downstream actors of CDKN2A and KRAS. They have shown that the SASP induced by the inhibition of CDK6/4 and MEK led to vascular remodelling that increased blood vessel density and permeability, making the tumour more sensitive to the cytotoxic chemotherapy [50]. It is well accepted that conventional RT induces senescence, and thus, could also lead to a beneficial SASP [132,133]. Protons and carbon ions can also trigger senescence [134,135,136]. A study performed on glioma cell lines irradiated with carbon ions showed that cells did not die of apoptosis or of autophagy but became senescent regardless of the p53 status of the cell line [134]. The role of senescent cells and associated SASP in cancer treatment can lead to possible benefit (vascular remodelling, immune cell attraction) or liability (promotion of cell proliferation and invasion) for tumour progression [137,138]. In PDAC, the work of Ruscetti et al. indicates that inducing senescence could be a promising approach to improve response to treatment.

Several studies have shown the implication of miRNAs and EVs in pancreatic tumour progression and the influence of low LET photon radiation on the EVs cargoes. However, very few data were recorded regarding the effect of charged particle irradiation on these elements in pancreatic cancer. However, Yu et al. recently demonstrated the effect of carbon ion irradiation on miRNAs expression transported by EVs in prostate cancer [139]. They compared the miRNAs extracted from exosomes derived from prostate cancer patient blood samples before and after carbon ion radiotherapy. The analysis evidenced an altered expression of 57 miRNAs present in the exosomes. In view of these results, we can suggest that charged particle irradiation might influence the EVs cargoes in pancreatic cancer as well.

Finally, the ability of PDAC cells to manage ROS suggests that direct DNA damaging radiation such as heavy charged particles would be more efficient than photons. In addition, the characteristic distribution of ROS after charged particles might affect the capacity of PDAC to handle these radio-induced ROS [126]. 

The peculiar distribution of DNA damage (or damage to other cellular component) and ROS production intensify cell response to charged particles compared to conventional photon irradiation. The alteration of pathways usually involved in resistance to treatment opens a window for higher LET ions to counteract these mechanisms and to extend the tumour local control, as summarised in Figure 1. 

### 4.3. Pushing Further PDAC Local Control: Charged Particles and Targeted Drug Combination

Enhanced DNA repair response to DNA damage is a cause of tumour resistance. Hence, we propose to further potentiate the effect of charged particles by targeting poly(ADP-ribose) polymerase (PARP) and homologous recombination (HR). 

Indeed, about 10% of PDAC patients present mutations in BRAC2 and ATM. Patients with BRCA1/2 mutations have shown beneficial responses to PARP inhibitors [140]. Upon DNA damage, PARP binds to single-strand breaks (SSBs), which, in turn, activates its catalytic domain, provoking the recruitment of other DNA damage repair proteins. PARP inhibitors, like Olaparib, prevent DNA damage repair to occur by stabilising the SSBs. These SSBs are in turn translated into DSBs at the replication fork. These DSBs, produced during the S phase, are repaired through the HR pathway for which intact BRCA1 and BRCA2 are required. The results of the phase III Pancreas Cancer Olaparib Ongoing (POLO) trial, which showed great promise for enrolled patients with metastatic pancreatic cancer with mutation in BRCA1 or BRCA2, who were previously treated with platinum-based chemotherapy. The enrolled patients were randomised to receive either Olaparib or placebo as the maintenance therapy. The study met its primary endpoint with a progression free survival (PFS) in the Olaparib group significantly longer compared to placebo. Furthermore, significant responses were seen in 20 patients in the treated group compared to six in the placebo group. However, no overall survival benefit could be recorded after analysis of 46% of events [141]. As shown in both ovarian and breast cancer, the POLO trial demonstrated the potential for PARP inhibition in pancreatic cancer, likely setting a new standard of care in patients presenting pancreatic cancer with germline BRCA1 or BRCA2 mutation. 

Recently, Szymonowicz et al. compared BRCA2-proficient BxPC3 and Capan-1 pancreatic cancer cells with BRCA2-deficiency, and showed that both cell lines were more sensitive to proton than to photon irradiation. The sensitising effect was even more noticeable in Capan-1 cells, leading the authors to suggest a predominant role of HR in the repair of clustered DNA damage induced by protons [142]. This was also demonstrated in lung cancer and glioblastoma cell lines where the impairment of HR led to a higher sensitisation after proton irradiation compared to photons for which the effect of NHEJ pathway inhibition was more pronounced [143].

In a previous work from our group, PARP and RAD51 inhibitors, Olaparib (AZD2281) and B02, respectively, were combined in order to sensitise cancer cells to proton irradiation [136]. Pancreatic cell lines, KP4 and PANC1, were radiosensitised to protons by each of the two inhibitors while the combination further increased cell death only for the fast cycling KP4 cell line. RAD51 is one of the key proteins of the HR pathway that is often overexpressed in cancer cells, notably in PDAC, and is thus now considered a clinically relevant target for combined therapies [144,145,146]. RAD51 inhibitors, such as B02, lead to HR inhibition, hence sensitising cells to DSBs [147,148,149,150,151]. This strategy could be used to counteract the resistance associated with PARP inhibitor but also to extend their use to patients without BRCA mutations. Furthermore, exposure to charged particles leads to the release of smaller DNA fragments compared to photons and notably fragments smaller than 40 base pairs [152]. Interestingly, Ku70/Ku80 heterodimer, a key player in non-homologous end joining (NHEJ) DSB repair pathway, is not able to bind these small fragments [152]. This means that the canonical NHEJ pathway is less efficient after irradiation with high LET particles. The repair of DSBs is thus in the hands of HR or alternative NHEJ pathways. This alternative NHEJ is dependent on PARP-1 [153]. Moreover, it has been shown that HR proteins such as RAD51 are downregulated under hypoxia [154,155]. These last observations show that combining PARP inhibitor with high LET particles such as carbon ions is very promising for improving local control. A combination of PARP inhibitor with a RAD51 inhibitor would push even further the local control obtained with charged particles irradiation [151,156]. 

## 5. PDAC Systemic management: Charged Particles to Trigger an Immune Response

In addition to the mechanisms cited above, immune evasion is also recognised to play a major role in tumour resistance. PDAC was originally considered as a ‘non-immunogenic’ neoplasm. However, it was recently demonstrated that PDAC immune microenvironment can play a considerable role in tumour evasion. 

### 5.1. Immune Evasion in PDAC: T Cells and Tumour-Associated Macrophages

T cells are abundant in the stroma of PDAC, and patients with higher levels of CD4+ and/or CD8+ T cells have significantly prolonged survival. Unfortunately, a large number of PDAC tumours demonstrate increased infiltration of T regulatory cells, myeloid-derived suppressor cells (MDSCs) and M2-like macrophages, blocking the activities of CD4+ and CD8+ T cells. PDAC is also characterised by an abundant desmoplastic stromal reaction restricting the infiltration of T cells into the tumour. T cell activation is determined by a balance between the signals of both co-stimulatory or co-inhibitory ligands and receptors. CTLA-4 is a CD28 homolog with a high affinity for B7-1 and B7-2 ligands. The interaction between CD28:B7-1/2 serves as a co-stimulatory signal for T cells, while the interaction between CTLA-4:B7-1/2 acts as a co-inhibitory signal to terminate the immune response. Thus, an effective way to promote lymphocyte activation is by reducing their inhibition. The checkpoint inhibitor Ipilimumab (anti-CTLA-4) has been administered as a single agent to patients with locally advanced or metastatic pancreatic cancer with disappointing results, as it was unable to demonstrate efficacy or prolong patient survival [157]. Another way to target T cell responses is through the PD-1/PD-L1 pathway. PD-1 is also a receptor of the CD28 family. This receptor can recognise two ligands: programmed death ligand 1 (PD-L1) and programmed death ligand 2 (PD-L2). The engagement of PD-1 by its ligands induces T cell inactivation. Anti-PD-1/PD-L1 have demonstrated clinical efficacy in several cancer types such as melanoma, non-small cell lung cancer and classical Hodgkin lymphoma. However, the same therapy showed no therapeutic benefits in patients with pancreatic cancer [158]. In their work, Ruscetti et al. combined a CDK4/6 inhibitor (palbociclib) to a MEK inhibitor (trametinib) in two mouse models of PDAC [50]. The SASP induced by this combination led to vascular remodelling with an increased permeability. They showed that exhausted CD8+ T cells easily infiltrated the tumour, as demonstrated by the presence of several exhaustion markers, PD-1, 2B4, CTLA-4 and LAG3, at the surface of the isolated cytotoxic T cells. CD8+ T cells isolated from tumours treated with senescence-inducing agents and anti-PD-1 combined therapy, displayed reduced expression of exhaustion markers as well as increased expression of several activation markers when compared to CD8+ T cells isolated from tumours treated with the senescence-inducing agents only [50].

Tumour-associated macrophages (TAMs) represent major actors in the tumour microenvironment. The presence of macrophages has been associated with poor prognosis in several cancer types and PDAC [159,160,161]. In the context of PDAC, TAMs were thought to originate only from circulating monocytes. However, it was recently shown that TAMs in pancreatic cancer can originate from circulating monocytes as well as from embryonic precursors [162]. Regardless of developmental origin, macrophages can adopt several functions, and are classified using the M1-M2 scale, similarly to the Th1-Th2 classification used for T cells. M1 macrophages, also known as classically activated, have pro-inflammatory and anti-tumour properties, whereas M2 macrophages, also known as alternative activated macrophages, exhibit anti-inflammatory and pro-tumour capabilities. In the tumour microenvironment, macrophages tend to adapt a M2-like phenotype, making them attractive targets for anti-tumour interventions. Strategies to target TAMs focus on their depletion, on blocking their recruitment into the tumour or on rewiring their phenotype towards the anti-tumour M1 phenotype. 

Trabectedin (ET-743), a molecule derived from sea squirt, *Ecteinascidia turbinate*, is approved for the treatment of advanced soft tissue sarcoma and ovarian cancers that relapsed. In addition to its anti-proliferating abilities, this drug is able to induce the depletion of myelomonocytic cells through the activation of caspase-8 via TNF-related apoptosis-inducing ligand (TRAIL) receptors [163]. A recent study, focused on the epigenetic profile of T cells after treatment with trabectedin, revealed that the depletion of TAMs can switch the phenotype of T cells from pro-tumour to anti-tumour in PDAC. Indeed, tumour-infiltrating lymphocytes (TILs) showed a reactivation both at the epigenetic and functional levels, with a switch from IL10-secreting T cells towards an effector/memory phenotype [164]. Combining trabectedin with checkpoint inhibitors might be efficient in PDAC patients, as it was successful in a murine model of ovarian cancer [165]. 

Circulating monocytes can be recruited into the tumour by the CCL2-CCR2 axis and the blockade of CCR2 decreases monocyte recruitment, tumour growth and metastasis in an orthotropic model of PDAC [166]. This pre-clinical study was followed by a phase Ib clinical trial testing CCR2 blockade in combination with chemotherapy in patients with advanced PDAC, which demonstrated that more patients achieved partial response when treated with CCR2 inhibitor and FOLFIRINOX [167]. Circulating monocytes are also recruited through the colony-stimulating factor 1 (CSF1)/CSF1 receptor (CSF1R) axis. It was shown that targeting macrophages with an inhibitor of CSF1R was able to increase mouse survival in the genetic KPC PDAC mouse model [168]. The inhibition of CSF1R can also improve the response of chemotherapy in an orthotopic mouse model [169]. A similar conclusion was drawn after treatment of PDAC bearing mice with RT and anti-CSF1 antibody [170]. Furthermore, the inhibition of CSF1/CSF1R is also capable of modulating the phenotype of macrophages in order to boost T cells [171]. A pilot study, NCT03153410, will be looking at the efficiency of cyclophosphamide, pembrolizumab, GVAX (pancreatic cancer vaccine) and IMC-CS4 (CSF1R monoclonal antibody) combinations in patients with BR-PDAC. 

Several approaches have been developed to switch the phenotype of M2 macrophages with pro-tumour abilities towards M1 macrophages with anti-tumour properties. CD40 is a member of the tumour necrosis factor (TNF) receptor superfamily and is expressed by immune cells such as B cells, dendritic cells (DC) and monocytes. In combination with gemcitabine therapy, CD40 agonists are capable of re-educating macrophages. When mice were treated with CD40 agonist, macrophages in the KPC tumours upregulated the expression of MHC class II and CD86, which is consistent with a M1 phenotype, when compared to untreated controls [172]. Nab-paclitaxel has also been demonstrated to induce anti-tumour immunity through the reprogramming of tumour-associated macrophages via TLR4 in vitro and in vivo. Indeed, treatment of RAW 264.7 cells with nab-paclitaxel induced an increased expression of IL-1alpha, IL-1beta, IL-6, IL-12p40 and TNF-alpha. The drug was also shown to induce M1 polarisation in an orthotopic murine model. Indeed, flow cytometry analysis revealed that nab-paclitaxel is able to increase the MHC II+ CD80+ CD86+ macrophage population [173,174]. In a similar fashion, lipopolysaccharide (LPS), known as an TLR4 agonist, can be used alone or in combination with IFN-γ to switch the polarisation of macrophages towards a M1 phenotype in order to induce anti-tumour response [175]. 

### 5.2. Conventional RT and Immune Response in PDAC 

It is well known that RT induces an immune response. Whether this response is in favour of immune stimulation or of immune repression is not trivial, as both types of responses are observed [176,177,178]. 

A combination of checkpoint inhibitors for RT should be efficient as RT upregulates PD-L1 expression in PDAC. Indeed, RT upregulates the expression of PD-L1 in KPC and Pan02 cell lines when compared to unirradiated control cells [179]. Anti-PD-L1 treatment significantly enhanced the tumour growth delay observed in vivo after giving high doses (12 Gy or 5 × 3 Gy) to Pan02 tumour model in mice. Several inflammatory cytokines were analysed in the sera of treated mice, stromal derived factor 1 (SDF-1) was significantly downregulated after receiving anti-PD-L1 and radiotherapy. Knowing that SDF-1 is implicated in the creation of an immunosuppressive microenvironment, the authors concluded that combining radiotherapy to checkpoint blockade is able to influence the tumour microenvironment and give favourable results [179]. This pre-clinical study demonstrated the efficacy of combining checkpoint blockers to RT, since checkpoint blockers on their own had failed to show benefits in PDAC patients. However, a recent study demonstrated that inhibiting autophagy might be able to sensitise PDAC tumours to dual immune checkpoint blockade (anti-PD-1 and anti-CTLA-4 antibodies) [180]. By screening a panel of human PDAC cell lines, the authors showed that MHC I was predominantly localised within the autophagosomes and lysosomes when compared to non-transformed human pancreatic ductal epithelial cells. Furthermore, the reduced expression of MHC I in the plasma membrane was shown to facilitate the evasion of cancer cells from cytotoxic T cells, which are capable of recognising tumour antigens when presented by MHC I. Inhibiting autophagy by the use of chloroquine increased MHC I expression levels in the plasma membrane in vitro as well as in vivo. However, as previously demonstrated, the use of chloroquine as a mono-therapy is unable to reduce the tumour burden [181], whereas combining chloroquine to dual immune checkpoint blockade significantly decreased tumour weight on treated mice bearing orthotopic tumours (HY15549), underlying the important role of autophagy to mount an immunological response [180]. The role of autophagy during pancreatic cancer development and progression is not as straightforward as mentioned above [46,182]. For example, autophagy has also been shown to play a major role in triggering immunological cell death (ICD) since it leads to ATP release which is one requirement for an efficient immune response. 

In order for conventional RT to activate an immune response, it has been demonstrated that the activation of cyclic GMP-AMP synthase (cGAS)/stimulator of interferon genes (STING) pathway is required. Indeed, the activation of the cGAS-STING pathway upregulates the expression of type I interferon, which plays an important role in the recruitment of DCs. However, it was shown that RT can upregulate the expression of Trex1, an exonuclease capable of degrading cytosolic dsDNA, thus preventing the activation of the cGAS-STING pathway. RT seems to upregulate the expression of Trex1 at doses around 12 to 18 Gy in different mouse and human carcinoma cell lines [183]. 

Regarding TAMs recruitment, the blockade of CCL2-CCR2 could also be used in combination with radiotherapy since CCL2 is responsible for radioresistance in a syngeneic mouse model of PDAC [184]. RT increased the expression of PDAC derived CCL2 leading to an increase in inflammatory monocytes/macrophages. In a surprising manner, the phenotype of these inflammatory monocytes/macrophages isolated from tumours after radiotherapy did not reveal differences in gene expression when compared to sham irradiated control. However, mice treated with a combination of RT and anti-CCL2 antibody showed prolonged survival probably due to impairment in inflammatory monocyte recruitment [184].

Finally, RT has the potential to switch TAMs phenotype from M1 to M2. For example, low-dose RT (2 Gy) has also been shown to be capable of inducing the reprogramming of tumour-associated macrophages in a genetic mouse model of insulinomas. Indeed, the expression of the M1 marker, iNOS as well as several M2 markers, namely, HIF-1, Ym-1, Fizz-1 and Arg1, was investigated in CD11b+ peritoneal macrophages from RT5 mice after receiving whole-body irradiation. Radiation led to an increase in iNOS expression, while reducing the expression of M2 markers in peritoneal macrophages. However, the authors observed that local irradiation alone is unable to swift the phenotype of TAMs toward a M1-like phenotype [185]. Furthermore, the same group also demonstrated that macrophages could be re-tuned towards an anti-tumour phenotype after whole body irradiation in RT5 mice [186]. These studies demonstrated the potential of RT to elicit anti-tumour immunity. However, it has to be noted that the effect of conventional RT is highly is dependent on the dose and the regimen schedule used [187]. 

### 5.3. Charged Particles to Improve Immune Response in PDAC

Charged particles feature an attracting dose deposition profile associated with a better dose conformation. This allows sparing circulating T cells and other immune cells during treatment. As explained in the review of Durante and Formenti, although the damages induced by charged particles to lymphocytes are more complex, it was shown in vivo that the size of the irradiated field is more relevant than DNA damage complexity to induce lymphopenia [188].

As evidenced in the previous section, charged particles also present an increased density of ionisation leading to an enhanced biological response due to the higher complexity of DNA damage. This enhanced response of tumour cells could help trigger T cells and TAMs response. 

For example, it was recently demonstrated that RT induces ICD in a dose-dependent manner and that particles could increase the immunogenic response [189,190]. Indeed, translocation of calreticulin to the cell membrane of irradiated cells was observed, as well as the release of high mobility group box 1 (HMGB-1) and ATP release. The release of these damage-associated molecular patterns (DAMPs) can trigger a cascade leading to the activation of DCs, which will prime CD8+ T cells. In vitro study has suggested that higher LET radiation may lead to a broader immunogenic response. Indeed, protons mediate calreticulin translocation to cell surface at higher levels than photons, leading to increased cross priming and higher sensitivity to cytotoxic T cells [190]. Furthermore, proton therapy did not increase the expression of PD-L1 on cancer cells, meaning that the activity of T cells would not be inhibited as it is the case for conventional RT. High-LET radiation was also shown to increase autophagy via the unfolded protein response (UPR), suggesting that high-LET radiation might effectively induce ICD and subsequent immune response in the context of pancreatic cancer [191,192]. A phase II clinical trial (NCT01494155) is currently underway to test the efficacy of capecitabine and hydroxychloroquine combined to photon or proton irradiation to control the growth of tumours in 50 patients with pancreatic cancer.

Particle therapy could help in the rewiring of TAMs phenotype. Indeed, protons were shown to partially reprogram, in vitro, M2-polarised macrophages towards an anti-tumour M1 phenotype [193]. Moreover, as opposed to photons, carbon ions were able to increase M1 and reduce M2 populations associated with increased abundance of cytotoxic CD8+ T cells in vivo in a mouse model of glioblastoma [122]. 

The role of FLASH PT to modulate the immune response is also worth investigating. Indeed, TGFβ1 level 24 h and 1 month after irradiation was significantly reduced normal human diploid lung fibroblasts with ultrahigh dose-rate irradiation compared to standard regimen [194]. Two other works mentioned the potential of FLASH PT and both evidenced the involvement of an immune response. Firstly, the effect of FLASH PT was evaluated on C57BL/6 mice 8 to 36 weeks after whole thorax irradiation and compared to standard dose-rate PT [195]. The authors reported a reduction in lung fibrosis and skin dermatitis. A genome wide microarray analysis revealed an elevation of DCs maturation, protein kinase C signalling in lymphocytes, TH1 pathway modulation and calcium-induced T lymphocyte apoptosis after standard regimen while they were decreased after FLASH dose-rate indicating the involvement of an immune response in the reduction of toxicities. The other study compared the efficacy of 18 Gy FLASH PT to standard dose-rate on the eradication of Lewis Cell Carcinoma syngeneic, orthotopic mouse model [196]. Mice irradiated with FLASH PT presented smaller tumours with an increased recruitment of CD4+ and CD8+ T cells in the tumour core than standard dose-rate irradiation. In their review, Dutt et al. assembled evidence on immunomodulation from novel dose-rate regimen such as FLASH RT [197]. For example, they presented that conventional irradiation led to an increased release of CCL2, which in turn attracted monocytes at tumour sites and promoted their differentiation into TAMs. Furthermore, HIF-1α, upregulated after RT, increased the expression of CSF1, which polarised these TAMs towards an M2 phenotype. M2 TAMs secretes TGFβ, which is known to convert CD4+ T cells into T regulatory cells [197]. In short, they showed that conventional RT can lead to an immunosuppressive response. However, this is not as simple as conventional RT is able to trigger both immunosuppressive and immunostimulatory responses [198]. 

The advantages of charged particles over photons on the immune response are summarised in Figure 1. 

Protons and carbon ions downregulate the expression of genes associated with an immunosuppressive response such as HIF-1α, VEGF, CCL2 or even TGFβ at ultrahigh dose-rate, but also lead to a stable expression of PD-L1. We can then expect charged particles to trigger an immunostimulatory response. Combining PARP and RAD51 inhibitor with charged particles could further enhance this response. Indeed, PARP inhibitors display an immunomodulatory effect by promoting the cGAS/STING pathway, thanks to the accumulation of cytosolic DNA, and by stimulating the expression of type I IFN, as explained in [199]. A combination with RAD51 inhibition would further push the STING pathway mediating an innate immune response by accumulation of cytosolic DNA [200]. 

Such a combination of charged particles with PARP and RAD51 inhibitor seems a promising approach to loco-regional control of PDAC with an increased level of clustered DNA whose repair relies mostly on HR and alt-NHEJ pathways. This combination could also lead to a systemic response by favouring an immune response. 

## 6. Conclusions

Current treatment modalities used for pancreatic cancer have not been able to dramatically change the course of this deadly disease. A variety of resistance mechanisms and induced toxicities of the current treatment have hampered their curative potential. Fortunately, the recent developments made in the field of radiotherapy allow highly conformal technique such as SBRT and charged particle therapy aiming at dose escalation and improved local control with limited toxicities to OARs. Several differences in tumour response to charged particles compared to photons have been highlighted in this review. These differences evidence that charged particles could help to counteract some of the resistance mechanisms found in PDAC. We suggest that charged particles also hold great promises when combined with DNA repair inhibitors such as PARP and RAD51 to improve local control. Moreover, this combination could be able to mount an efficient immune response thanks to the release of several DAMPs enabling the activation of T cells and downregulation of immunosuppressive genes. Pre-clinical studies also demonstrate that unlike conventional radiotherapy, proton and carbon ions are able to reprogram macrophages towards an anti-tumour phenotype. Additionally, senescence-associated therapies, such as radiotherapy, can create several vulnerabilities in pancreatic cancer leading to drug delivery enhancement and T cell infiltration. Therefore, combining particle therapy to complementary DNA repair inhibitor could be beneficiary in the context of pancreatic cancer in order to increase local and distant control of the tumour.

The availability as well as the cost of this technique have, up to now, limited its use in clinical practise over conventional photon radiotherapy. Coming clinical trials have to evidence the balance between efficiency and cost to help address which patients would benefit from particle therapy. Currently, the clinical data regarding the advantages of particle therapy over conventional radiotherapy are limited, but the results presented above are encouraging. With several clinical trials underway, the effects of particle therapy in the context of pancreatic cancer should bring forward this promising form of treatment.

## Figures and Tables

**Figure 1 ijms-21-04767-f001:**
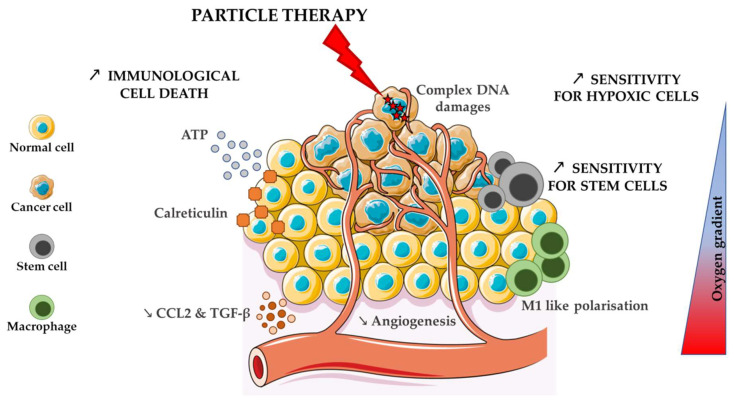
Attractive effects of charged particles in comparison to photons on the different tumour cell types.

**Table 1 ijms-21-04767-t001:** Ongoing clinical trials for pancreatic cancer patients treated with protons or carbon ions.

ID	Condition	Year	Status	*N*	Radiation	Dose	Concurrent Chemotherapy
NCT03885284	R-PDAC (adjuvant)	2019	Recruiting	12	Protons	25 Gy (RBE) in five fractions	mFOLFIRINOX
NCT01591733	R-PDAC (neoadjuvant)	2012	Active	48	Protons (Photons)	25 Gy (RBE) in five fractions (30 Gy in 10 fractions)	FOLFIRINOX + Capecitabine
NCT01494155	R-PDAC (neoadjvant)	2011	Active	50	Protons (Photons)	25 Gy (RBE) in five fractions (30 Gy in 10 fractions)	Capecitabine + Hydroxychloroquine
NCT03822936	R-PDAC (neoadjuvant)	2019	Recruiting	30	Carbon ions	38.4 Gy (RBE) in eight fractions	N/A
NCT02598349	Unresectable	2015	Recruiting	60	Protons	63 Gy (RBE) in 28 fractions	Capecitabine
NCT04194268	Unresectable	2019	Recruiting	25	Carbon ions	48 Gy (RBE) in 12 fractions	N/A
NCT03652428	LA-PDAC	2018	Recruiting	24	Protons	75 Gy (RBE) in 15 fractions	Gemcitabine
NCT03652428	LA-PDAC	2018	Recruiting	24	Protons	75 Gy (RBE)E in 15 fractions	Nab-paclitaxel + Gemcitabine
NCT04082455	LA-PDAC	2019	Recruiting	49	Carbon ions	60-67.5Gy (RBE) in 15 fractions	N/A
NCT01821729	LA-PDAC	2013	Active	50	Protons (Photons)	25 Gy (RBE) in five fractions (if persistent vascular involvement 50.4 Gy with vascular boost to 58.8 Gy)	FOLFIRIRINOX + Losartan
NCT03536182	LA-PDAC	2018	Active	110	Carbon ions (Photons)	55.2 or 57.6 Gy (RBE) in 12 fractions (50.4–56 Gy in 28 fractions)	• Gemcitabine • Gemcitabine + Nab-paclitaxel • FOLFIRINOX

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
