# Peer review of "Could Protons and Carbon Ions Be the Silver Bullets Against Pancreatic Cancer?"

_ijms, 2020, doi:10.3390/ijms21134767_

Round 1
Reviewer 1 Report
The review covers a very important topic for the treatment of pancreatic cancer with Proton vs. Photon therapy and discusses the relevant information on this topic well. I very much support the publication of this great written review after a major revision including the points I have mentioned below.
- Please include a more detailed section describing other relevant modes of tumor hypoxia within a solid tumor (e.g. severe hypoxia, chronic cycling hypoxia) to the “Mechanisms of Resistance”-Section explaining its documented effects on radiosensitivity and acquired radioresistance and discuss these effects in the view of proton vs. photon therapy.
- Please include and discuss recent work on the necessity for DNA repair pathways upon Photon and Proton irradiation in DNA repair deficient pancreatic cell lines (e.g. Szymonowicz K, Krysztofiak A, Linden JV, et al. Proton Irradiation Increases the Necessity for Homologous Recombination Repair Along with the Indispensability of Non-Homologous End Joining. Cells. 2020;9(4):889. Published 2020 Apr 5. doi:10.3390/cells9040889doi: 10.3390/cells9040889)
Reviewer 2 Report
The paper is a comprehensive elaboration on molecular mechnisms that might be responsible for treatment effects in pancreatic cancer patients.
The authors suggest heavy ions (alone or combined with chemotherapy) a good radiotherpeutic option due to several mechanisms including heavy ions influence on DNA damage, tumour microenvironment and immune response.
Giving a broad spectrum of data on experimental studies the authors present their own conclusion on probable mechanisms that might be used in arranging an effective combined (R+CT) treatment.
This valuable paper does not however convince a reader that the data presented do indicate heavy ions and even more protons to be the best radiotherapy method for pancreatic cancer patients.The paper title does not fully reflects the content of the article.
It would be worth to compare radiobiological features of charged particles (as an experimental, expensive and little available method) to a hypofractionated highly conformal photons (SBRT) i.e. in terms of their influence on tumour microenvironment or immune response.
I would also suggest to add to the impressive discussion on molecular mechanisms a discussion on classic radiobiological features of pancreatic cancer (i.e. its relatively low alpha/beta ratio around 4-5 Gy, hypoxia contribution) which are of great importance in choosing treatment options.
Round 2
Reviewer 1 Report
The review covers a very important topic for the treatment of pancreatic cancer with Proton vs. Photon therapy and discusses the relevant information on this topic well. I very much support the publication of the revised manuscript.